# Gradient-free variational learning with conditional mixture networks

**Conor Heins**[*][†][1,2]
conor.heins@verses.ai

**Hao Wu**[†][1]
wuhaomxhy@gmail.com

**Dimitrije Markovic**[†][1,3]
dimitrije.markovic@tu-dresden.de

**Alexander Tschantz**[1,4]
alec.tschantz@verses.ai

**Jeff Beck**[‡][1,5]
jeff.beck@duke.edu

**Christopher L. Buckley**[‡][1,4]
christopher.buckley@verses.ai

[1] VERSES
Los Angeles, CA, USA

[2] Max Planck Institute of Animal Behavior
Department of Collective Behaviour
Konstanz, Germany

[3]Chair of Cognitive Computational Neuroscience
Technische Universität Dresden
Dresden, Germany

[4] School of Engineering and Informatics
University of Sussex
Brighton, UK

[5] Department of Neurobiology
Duke University
Durham, NC, USA

## Abstract

Balancing computational efficiency with robust predictive performance is cru-
cial in supervised learning, especially for safety-critical applications. While deep
learning models are accurate and scalable, they often lack calibrated predictions
and uncertainty quantification. Bayesian methods address these issues but are of-
ten computationally expensive. We introduce CAVI-CMN, a fast, gradient-free
variational method for training conditional mixture networks (CMNs), a prob-
abilistic variant of the mixture-of-experts (MoE) model. Using conjugate priors
and Pólya-Gamma augmentation, we derive efficient updates via coordinate ascent
variational inference (CAVI). We apply this method to train conditional mixture
networks on classification tasks from the UCI repository. CAVI-CMN achieves
competitive and often superior predictive accuracy compared to backpropagation
(i.e., maximum likelihood estimation) while maintaining posterior distributions
over model parameters. Moreover, computation time scales in model complex-
ity competitively to both MLE and other gradient-based solutions like black-box
variational inference (BBVI), while running overall much faster than BBVI and
sampling-based inference and with similar speed to MLE. This combination of
probabilistic robustness and computational efficiency positions CAVI-CMN as a

---

[*]Corresponding author

[†]Co-first authors

[‡]Co-senior authors

Workshop on Bayesian Decision-making and Uncertainty, 38th Conference on Neural Information Processing
Systems (NeurIPS 2024).

building block for constructing discriminative models that are fast, gradient-free, and Bayesian.

# 1 Introduction

Modern machine learning methods attempt to learn functions of complex data (e.g., images, audio, text) to predict information associated with that data, such as discrete labels [Bernardo et al., 2007]. Deep neural networks (DNNs) have demonstrated success in this domain, owing to their universal function approximation properties [Park and Sandberg, 1991] and scalable optimization algorithms for training them [Amari, 1993]. Despite their performance and scalability, DNNs do not provide well calibrated predictions and uncertainty estimates [Wang et al., 2021, Shao et al., 2020]. This limits the applicability and reliability of using DNNs in safety-critical applications like autonomous driving, medicine, and disaster response [Papamarkou et al., 2024].

Here we introduce a gradient-free variational learning algorithm for a probabilistic variant of a two-layer, feedforward neural network — the conditional mixture network or CMN — and measure its performance on supervised learning benchmarks. This method rests on coordinate ascent variational inference (CAVI) [Wainwright et al., 2008, Hoffman et al., 2013] and hence we name it CAVI-CMN. CAVI-CMN maintains the predictive accuracy and scalability of an architecture-matched feedforward neural network fit with maximum likelihood estimation, while maintaining full distributions over its parameters and generating calibrated predictions, as measured in relationship to state-of-the-art Bayesian methods like the No U-Turn Sampler (NUTS) algorithm for Hamiltonian Monte Carlo [Hoffman et al., 2014] and black-box variational inference [Ranganath et al., 2014].

We summarize the contributions of this work below:

- Introduce and derive a coordinate ascent variational inference scheme for the conditional mixture network, which we term CAVI-CMN.

- CAVI-CMN matches, and sometimes exceeds, the performance of maximum likelihood estimation (MLE) in terms of predictive accuracy, while maintaining probabilistic benefits like high log predictive density and low calibration error. This is shown across a suite of 8 different supervised classification tasks.

- CAVI-CMN requires drastically less time to converge and overall runtime than the other state-of-the-art Bayesian methods like NUTS and BBVI.

# 2 Methods

Here, we introduce a variant of the Mixture-of-Experts (MoE) model [Jacobs et al., 1991] that makes its parameters amenable to gradient-free Bayesian learning. Jacobs et al. [1991] originally introduced MoEs as a way to improve the performance of neural networks by combining the strengths of multiple specialized models [Gormley and Frühwirth-Schnatter, 2019]. Non-Bayesian approaches to MoE typically rely on maximum likelihood estimation (MLE) [Jacobs et al., 1991], which can suffer from overfitting and poor generalization due to the lack of regularization mechanisms [Bishop and Svenskn, 2003].

The approach we propose, CAVI-CMN takes advantage of the conditional conjugacy of a mixture of linear experts, along with Pólya-Gamma (PG) augmentation [Polson et al., 2013] for the softmax layers, to make all parameters amenable to variational Bayesian inference. We use coordinate ascent variational inference (CAVI) to obtain posteriors over the weights of both the individual experts and the gating network [Bishop and Nasrabadi, 2006, Blei et al., 2017], without resorting to costly gradient or sampling computations.

## 2.1 The conditional mixture network

The conditional mixture network maps from a continuous input vector $\boldsymbol{x}_0 \in \mathbb{R}^d$ to its label $y \in \{1, \dots, L\}$. This is achieved with two layers: a conditional mixture of linear experts, which outputs a joint continuous-discrete latent $\left(\boldsymbol{x}_1 \in \mathbb{R}^h, z_1 \in \{1, \dots, K\}\right)$ and a multinomial logistic regression or softmax layer, which maps from the continuous latent $\boldsymbol{x}_1$ to the corresponding label $y$. Given a dataset of input-label pairs $(\boldsymbol{X}_0, Y) = \{\boldsymbol{x}_0^n, y^n\}_{n=1}^N$, the CMN defines a joint distribution over labels $Y$, latents $\boldsymbol{X}_1, Z_1$, and parameters $\boldsymbol{\Theta}$:

$$p(\boldsymbol{Y}, \boldsymbol{X}_1, \boldsymbol{Z}_1, \boldsymbol{\Theta} | \boldsymbol{X}_0) = p(\boldsymbol{\Theta}) \prod_{n=1}^{N} p_{\boldsymbol{\beta}_1}(y^n | \boldsymbol{x}_1) \, p_{\boldsymbol{\lambda}_1}(\boldsymbol{x}_1^n | \boldsymbol{x}_0^n, z_1^n) \, p_{\boldsymbol{\beta}_0}(z_1^n | \boldsymbol{x}_0^n) \tag{1}$$

$$p(\boldsymbol{\Theta}) = p(\boldsymbol{\beta}_1) \, p(\boldsymbol{\beta}_0) \, p(\boldsymbol{\lambda}_1)$$

where $p_{\boldsymbol{\lambda}_1}(\boldsymbol{x}_1^n | \boldsymbol{x}_0^n, z_1^n)$ refers to a mixture of linear models with parameters $\boldsymbol{\lambda}_1 = \boldsymbol{A}_{1:K}, \boldsymbol{\Sigma}_{1:K}^{-1}$, while $p_{\boldsymbol{\beta}_0}(z_1^n | \boldsymbol{x}_0^n)$ is a multinomial logistic (softmax) layer that outputs a probability over discrete latent $z_1^n$, which selects which of $K$ experts to use in predicting $\boldsymbol{x}_1^n$. $p_{\boldsymbol{\beta}_1}(y^n | \boldsymbol{x}_1)$ parameterizes a final (softmax) likelihood over the label $y^n$. A Bayesian network representation of the two layer CMN architecture is shown in Figure 1.

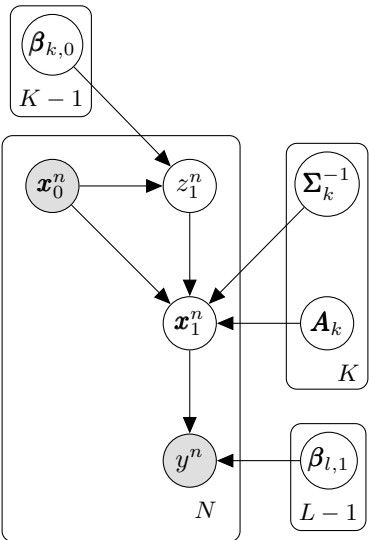

Figure 1: A Bayesian network representation of the two-layer conditional mixture network, with an input-output pair $\boldsymbol{x}_0^n, y^n$ and latent variables $\boldsymbol{x}_1^n, z_1^n$. Observations are shaded nodes, while latents and parameters are transparent.

## 2.2 Coordinate ascent variational inference with conjugate priors

In this section we summarize a variational approach for inverting the probabilistic model described in Equation (1). We posit the following approximate posterior over latents and parameters:

$$p(\boldsymbol{X}_1, \boldsymbol{Z}_1, \boldsymbol{\Theta} | Y, \boldsymbol{X}) \approx q(\boldsymbol{\Theta}) \prod_{n=1}^{N} q(z_1^n) \, q(\boldsymbol{x}_1^n | z_1^n) \qquad q(\boldsymbol{\Theta}) = q(\boldsymbol{\beta}_1) \, q(\boldsymbol{\beta}_0) \, q(\boldsymbol{\lambda}_1) \tag{2}$$

where $q(\boldsymbol{x}_1^n, z_1^n)$ corresponds to an approximate posterior over continuous $\boldsymbol{x}_1$ and discrete $z_1$ latent variables.

The mean-field factorized form of the approximate posterior [Svensén, 2003], combined with conjugate priors over the parameters $\boldsymbol{\Theta}$ (see Appendix A for their form), allows us to derive conditionally-conjugate updates for $q(\boldsymbol{X}_1, \boldsymbol{Z}_1)$ and $q(\boldsymbol{\Theta})$. We use an iterative update scheme for the parameters of the approximate posterior, often referred to as variational Bayesian expectation maximisation (VBEM) [Beal, 2003] or coordinate ascent variational inference (CAVI) [Bishop and Nasrabadi, 2006]. This consists in alternating updates to the posterior over latents and the posterior over parameters, split into a variational E-step and a variational M-step. Each step maximizes the evidence lower bound (ELBO), conditioned on the current setting of the other factor (i.e., $q(\boldsymbol{X}_1, \boldsymbol{Z}_1)$ or $q(\boldsymbol{\Theta})$).

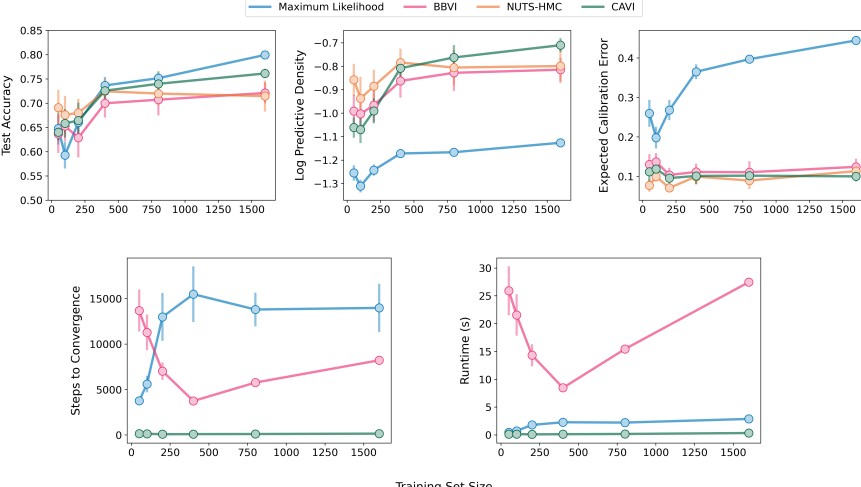

Figure 2: Performance and runtime results of the different inference algorithms on the 'Pinwheel' dataset from Johnson et al. [2016]. The standard deviation (vertical lines) of the performance metric is depicted together with the mean estimate (circles) over different model initializations. The top row of subplots show test accuracy (top left); log predictive density (top center), and expected calibration error (top right) as a function of training set size. The bottom row shows runtime metrics as a function of increasing training set size: the number of iterations required to achieve convergence (lower left); and the total runtime (in seconds, lower right). See Appendix G for details on run-time metrics.

Update to latents ('E-step')

$$q_t\left(\boldsymbol{x}_1^n, z_1^n\right) \propto \exp\left\{\mathbb{E}_{q_{t-1}(\boldsymbol{\Theta})}\left[\ln p_{\boldsymbol{\Theta}}(y^n, \boldsymbol{x}_1^n, z_1^n | \boldsymbol{x}_0^n)\right]\right\}$$

Update to parameters ('M-step')

$$q_t\left(\boldsymbol{\Theta}\right) \propto \exp\left\{\mathbb{E}_{q_{t-1}\left(\boldsymbol{x}_1^n, z_1^n\right)}\left[\ln p_{\boldsymbol{\Theta}}(y^n, \boldsymbol{x}_1^n, z_1^n | \boldsymbol{x}_0^n)\right]\right\}$$

$$(3)$$

The functional forms of these equations and the PG augmentation scheme needed to turn them into conditionally-conjugate updates, are given in detail in Appendix A and Appendix B.

## 3 Results

We fit CAVI-CMN on several real and synthetic datasets and compared it to three alternative inference methods for fitting the parameters of the CMN:

**MLE** — We obtained point estimates for the parameters $\boldsymbol{\Theta}$ of the CMN using maximum-likelihood estimation (backpropagation to minimize the negative log likelihood).

**NUTS-HMC** — The No-U-Turn Sampler (NUTS), an extension to Hamiltonian Monte Carlo (HMC) that incorporates adaptive step sizes [Hoffman et al., 2014]. This provides samples from a posterior distribution over $\boldsymbol{\Theta}$.

**BBVI** — Black-Box Variational Inference (BBVI) method [Ranganath et al., 2014]. BBVI maximizes the evidence lower bound (ELBO) using stochastic estimation of its gradients with respect to variational parameters.

Appendix C contains details of the hyperparameters used for each inference algorithm.

### 3.1 Predictive performance and efficiency

We fit all the inference algorithms on the Pinwheels dataset [Johnson et al., 2016] and 7 datasets from the UCI Machine Learning repository [Kelly et al., 2024]. The upper row of Figure 2 visualizes three

different performance metrics for the Pinwheels dataset as a function of the size of the training set. The CAVI-based approach achieves competitive test accuracy to MLE, as well as comparable log predictive density (LPD) and expected calibration error (ECE) to the other two Bayesian methods; all three Bayesian approaches outperform maximum likelihood estimation in LPD and ECE. This finding holds for 6 of the 7 datasets we tested (see Appendix E), and also holds across training set sizes, indicating robust sample efficiency and calibration. To further study the probabilistic performance of CAVI-CMN, we computed the widely applicable information criterion (WAIC), an approximate estimate of leave-one-out cross-validation [Vehtari et al., 2017, Watanabe and Opper, 2010]. Table 1 shows the WAIC scores for all methods evaluated on 7 UCI datasets. The CAVI-CMN approach consistently provided higher WAIC scores compared to the MLE algorithm, and WAIC scores that were on par with BBVI and NUTS.

The bottom row of Figure 2 shows that across training set sizes, all three gradient-based algorithms [4] exhibit an increase in runtime as the number of training data increases (which also scales the number of parameters for BBVI and CAVI). However the rate of increase varies significantly across different algorithms, with CAVI-CMN approach showing the best scaling behavior, both in terms of steps-to-convergence and absolute runtime. CAVI-CMN's runtime also scales competitively with MLE and BBVI along two other dimensions of model complexity: input dimension $d$ and number of expert learners $K$ (see Appendix F).

Thus, CAVI-CMN retains the probabilistic benefits of state-of-the-art Bayesian methods, as measured by metrics like test accuracy, LPD, and ECE, while also offering substantial advantages in terms of computational efficiency.

|      | Rice | Breast Cancer | Waveform | Vehicle Silh. | Banknote | Sonar | Iris |
|------|------|---------------|----------|---------------|----------|-------|------|
| CAVI | -0.1820 | -0.0504 | **-0.2921** | **-0.3281** | -0.0206 | -0.1544 | -0.0747 |
| MLE  | -0.3599 | -0.3133 | -0.5759 | -0.7437 | -0.3133 | -0.3133 | -0.5514 |
| NUTS | **-0.1278** | **-0.0324** | -0.3753 | -0.3767 | **-0.0110** | **-0.0306** | **-0.0413** |
| BBVI | -0.1739 | -0.0763 | -0.3618 | -0.4154 | -0.0382 | -0.0583 | -0.1544 |

Table 1: Comparison of widely-applicable information criterion (WAIC) for different methods evaluated on 7 different UCI datasets. The highest WAIC score for each dataset is highlighted in bold-face.

## 4 Conclusion

We introduced CAVI-CMN, a computationally efficient Bayesian approach for conditional mixture networks (CMN) that outperforms maximum likelihood estimation (MLE) in terms of predictive performance and calibration, as measured by LPD and ECE, and is competitive in terms of test accuracy on held out data. CAVI-CMN offers significant computational advantages over other Bayesian methods like Black-Box Variational Inference (BBVI) and the No-U-Turn Sampler (NUTS). While NUTS excels in inference quality, its computational cost is prohibitive for complex models. BBVI, though efficient, converges slower and has overall slower run-time than CAVI when applied to the CMN model.

The benchmark results demonstrate that CAVI-CMN matches the performance of BBVI and NUTS in terms of predictive accuracy, log-predictive density, and expected calibration error, while being considerably faster. The conjugate-exponential form of CAVI-CMN also makes it amenable to online learning with mini-batches of data, suggesting the extension of CAVI-CMN to deeper architectures and larger datasets. Overall, CAVI-CMN presents a promising tool for building fast, gradient-free and scalable Bayesian machine learning models.

**Acknowledgments and Disclosure of Funding**

We thank the anonymous reviewers for their valuable feedback. The authors would like to thank the members of the VERSES Machine Learning Foundations and Intelligent Systems groups for critical discussions and feedback that improved the quality of this work, with special thanks to Tommaso Salvatori, Tim Verbelen, Magnus Koudahl, Toon Van de Maele, Hampus Linander, and Karl Friston.

---

[4]We excluded NUTS from this analysis because its runtime was in general always orders of magnitude larger than the other methods

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

## Code availability

The code for using CAVI and the other 3 methods to fit the CMN model on the pinwheel and UCI datasets is available on the 'cavi-cmn' repository, hosted on the VersesTech GitHub organization: https://github.com/VersesTech/cavi-cmn.

## A    Coordinate ascent variational inference for conditional mixture networks

In this section we detail a variational approach for inverting the probabilistic model described in Equation (1) and computing an approximate posterior over latents and parameters specified as

$$p\left(\boldsymbol{X}_1,\boldsymbol{Z}_1,\boldsymbol{\Theta}|Y,\boldsymbol{X}\right) = \frac{p\left(Y,\boldsymbol{X}_1,\boldsymbol{Z}_1,\boldsymbol{\Theta},\boldsymbol{X}\right)}{p\left(Y|\boldsymbol{X}\right)} \approx q\left(\boldsymbol{\Theta}\right)\prod_{n=1}^{N} q\left(z_1^n\right)q\left(\boldsymbol{x}_1^n|z_1^n\right) \qquad (4)$$

where $q\left(\boldsymbol{x}_1^n|z_1^n\right)$ corresponds to a component specific multivariate normal distribution, and $q\left(z_1^n\right)$ to a multinomial distribution. Importantly, the approximate posterior over parameters $q\left(\boldsymbol{\Theta}\right)$ further factorizes [Svensén, 2003] as

$$q\left(\boldsymbol{\Theta}\right) = \prod_{l=1}^{L-1} q\left(\boldsymbol{\beta}_{l,1}\right) \prod_{k=1}^{K-1} q\left(\boldsymbol{\beta}_{k,0}\right) \underbrace{\prod_{j=1}^{K} q\left(\boldsymbol{A}_j,\boldsymbol{\Sigma}_j^{-1}\right)}_{=q(\boldsymbol{\lambda}_1)}$$

$$\begin{aligned} q\left(\boldsymbol{\beta}_{l,1}\right) &= \mathcal{N}\left(\boldsymbol{\beta}_{l,1};\boldsymbol{\mu}_{l,1},\boldsymbol{\Sigma}_{l,1}\right) \\ q\left(\boldsymbol{\beta}_{k,0}\right) &= \mathcal{N}\left(\boldsymbol{\beta}_{k,0};\boldsymbol{\mu}_{k,0},\boldsymbol{\Sigma}_{k,0}\right) \\ q\left(\boldsymbol{A}_j|\boldsymbol{\Sigma}_j^{-1}\right) &= \mathcal{MN}\left(\boldsymbol{A}_j;\boldsymbol{M}_j,\boldsymbol{\Sigma}_j,\boldsymbol{V}_j\right) \\ q\left(\boldsymbol{\Sigma}_j^{-1}\right) &= \prod_{i=1}^{h} \Gamma\left(\sigma_{i,j}^{-2};a_j,b_{i,j}\right) \end{aligned} \qquad (5)$$

We use the following conjugate priors for the parameters of the linear experts $\boldsymbol{\lambda}_1 = \left(\boldsymbol{A}_{1:K},\boldsymbol{\Sigma}_{1:K}^{-1}\right)$ and the regression coefficients $\boldsymbol{\beta}_0,\boldsymbol{\beta}_1$:

$$\begin{aligned} p\left(\boldsymbol{A}_k|\boldsymbol{\Sigma}_k^{-1}\right) &= \mathcal{MN}\left(\boldsymbol{A}_k;\boldsymbol{0},\boldsymbol{\Sigma}_k,v_0\boldsymbol{I}_{d+1}\right) \\ p\left(\boldsymbol{\Sigma}_k^{-1} \equiv \mathrm{diag}\left(\boldsymbol{\sigma}_k^{-2}\right)\right) &= \prod_{i=1}^{h} \Gamma\left(\sigma_{k,i}^{-2};a_0,b_0\right) \\ p\left(\boldsymbol{\beta}_{k,0}\right) &= \mathcal{N}\left(\boldsymbol{\beta}_{k,0};\boldsymbol{0},\sigma_0^2\boldsymbol{I}_{d+1}\right) \\ p\left(\boldsymbol{\beta}_{l,1}\right) &= \mathcal{N}\left(\boldsymbol{\beta}_{l,1};\boldsymbol{0},\sigma_1^2\boldsymbol{I}_{h+1}\right) \end{aligned} \qquad (6)$$

The above form of the approximate posterior, in combination with the conjugate priors in Equation (6), allows us to define tractable conditionally conjugate updates for each factor. This becomes evident from the following expression for the evidence lower bound (ELBO) on the marginal log likelihood

$$\mathcal{L}(q) = \mathbb{E}_{q(\boldsymbol{X}_1,\boldsymbol{Z}_1)q(\boldsymbol{\Theta})}\left[\sum_{n=1}^{N} \ln \frac{p_{\boldsymbol{\Theta}}\left(y^n,\boldsymbol{x}_1^n,z_1^n|\boldsymbol{x}_0^n\right)}{q\left(z_1^n\right)q\left(\boldsymbol{x}_1^n|z_1^n\right)}\right] + \mathbb{E}_{q(\boldsymbol{\Theta})}\left[\ln \frac{p\left(\boldsymbol{\beta}_1\right)p\left(\boldsymbol{\beta}_0\right)p\left(\boldsymbol{\lambda}_1\right)}{q\left(\boldsymbol{\beta}_1\right)q\left(\boldsymbol{\beta}_0\right)q\left(\boldsymbol{\lambda}_1\right)}\right] \qquad (7)$$

We maximize the ELBO using an iterative update scheme for the parameters of the approximate posterior, often referred to as variational Bayesian expectation maximisation (VBEM) [Beal, 2003] or coordinate ascent variational inference (CAVI) [Bishop and Nasrabadi, 2006, Blei et al., 2017]. The procedure consists of two parts:

First, we fix the posterior over the parameters (to randomly initialized values). Given the posterior over parameters, we update the posterior over latent variables (variational E-step) as

$$q_t\left(\boldsymbol{x}_1^n|z_1^n\right) \propto \exp\left\{\mathbb{E}_{q_{t-1}(\boldsymbol{\beta}_1)q_{t-1}(\boldsymbol{\lambda}_1)}\left[\ln p_{\boldsymbol{\beta}_1}\left(y^n|\boldsymbol{x}_1^n\right) + \ln p_{\boldsymbol{\lambda}_1}\left(\boldsymbol{x}_1^n|\boldsymbol{x}_0^n, z_1^n\right)\right]\right\}$$

$$q_t\left(z_1^n\right) \propto \exp\left\{\mathbb{E}_{q_{t-1}(\boldsymbol{\Theta})}\left[\left\langle\ln p_{\boldsymbol{\beta}_1,\boldsymbol{\lambda}_1}\left(y^n, \boldsymbol{x}_1^n|\boldsymbol{x}_0^n, z_1^n\right)\right\rangle_{q_t(\boldsymbol{x}_1^n|z_1^n)} + \ln p_{\boldsymbol{\beta}_0}\left(z_1^n|\boldsymbol{x}_0^n\right)\right]\right\}$$

(8)

Second, the posterior over latents that was updated in the E-step, is used to update the posterior over parameters (variational M-step) as

$$q_t\left(\boldsymbol{\beta}_1\right) \propto \exp\left\{\sum_{n=1}^{N}\mathbb{E}_{q_t(\boldsymbol{x}_1^n,z_1^n)}\left[\ln p_{\boldsymbol{\beta}_1}\left(y^n|\boldsymbol{x}_1^n\right)\right]\right\}$$

$$q_t\left(\boldsymbol{\beta}_0\right) \propto \exp\left\{\sum_{n=1}^{N}\mathbb{E}_{q_t(z_1^n)}\left[\ln p_{\boldsymbol{\beta}_1}\left(z_1^n|\boldsymbol{x}_0^n\right)\right]\right\}$$

$$q_t\left(\boldsymbol{\lambda}_1\right) \propto \exp\left\{\sum_{n=1}^{N}\mathbb{E}_{q_t(\boldsymbol{x}_1^n,z_1^n)}\left[\ln p_{\boldsymbol{\lambda}_1}\left(\boldsymbol{x}_1^n|z_1^n, \boldsymbol{x}_0^n\right)\right]\right\}$$

(9)

In the variational inference literature, the distinction between latents and parameters is often described in terms of 'local' vs 'global' latent variables [Hoffman et al., 2013], where local variables are datapoint-specific, and global variables are shared across datapoints. To detail the form of the updates to the parameters of the linear experts in Equation (9), i.e. $q_t(\boldsymbol{\lambda}_1) = q_t(\boldsymbol{A}_{1:K}, \boldsymbol{\Sigma}_{1:K}^{-1})$, first we note the form of the approximate posteriors over the latent variables $q(\boldsymbol{X}_1, Z_1)$:

$$q\left(\boldsymbol{X}_1|Z_1\right) = \prod_{n=1}^{N}\prod_{k=1}^{K}\mathcal{N}(\boldsymbol{x}_1^n; \boldsymbol{\mu}_{k,1}^n, \boldsymbol{\Sigma}_{k,1}^n)$$

$$q\left(Z_1\right) = \prod_{n=1}^{N}\text{Cat}(z_1^n; \boldsymbol{\gamma}^n)$$

(10)

The update to the $k^{\text{th}}$ expert's parameters $q(\boldsymbol{A}_k, \boldsymbol{\Sigma}_k^{-1})$ can written in terms of weighted updates to the Matrix Normal Gamma's canonical parameters $\boldsymbol{M}_k, \boldsymbol{V}_k, a_k$ and $b_k$, where the weights are provided by the sufficient statistics of $\{q\left(\boldsymbol{x}_1^1|z_1^1=k\right), q\left(\boldsymbol{x}_1^2|z_1^2=k\right), \dots, q\left(\boldsymbol{x}_1^N|z_1^N=k\right)\}$:

$$\boldsymbol{V}_k^{-1} = \boldsymbol{V}_{k,0}^{-1} + \sum_{n=1}^{N}\gamma_k^n\boldsymbol{x}_0^n\left(\boldsymbol{x}_0^n\right)^{\top}$$

$$\boldsymbol{M}_k = \left(\boldsymbol{M}_{k,0}\boldsymbol{V}_{k,0}^{-1} + \sum_{n=1}^{N}\gamma_k^n\boldsymbol{\mu}_{k,1}^n\left(\boldsymbol{x}_0^n\right)^{\top}\right)\boldsymbol{V}_k$$

$$a_k = a_{k,0} + \frac{\sum_{n=1}^{N}\gamma_k^n}{2}$$

$$b_{i,k} = b_{i,k,0} + \frac{1}{2}\left(\sum_{n=1}^{N}\gamma_k^n\left[\boldsymbol{\Sigma}_{k,1}^n + \boldsymbol{\mu}_{k,1}^n(\boldsymbol{\mu}_{k,1}^n)^{\top}\right]_{ii} - \left[\boldsymbol{M}_k\boldsymbol{V}_k^{-1}\boldsymbol{M}_k^T\right]_{ii} + \left[\boldsymbol{M}_{k,0}\boldsymbol{V}_{k,0}^{-1}\boldsymbol{M}_{k,0}^T\right]_{ii}\right)$$

(11)

where the notation $[\cdot]_{ii}$ selects the $i^{\text{th}}$ element of the diagonal of the matrix in the brackets.

However, the update equations described in Equation (8) and in the first two lines of Equation (9) for $q(\boldsymbol{\beta}_0), q(\boldsymbol{\beta}_1)$ are not computationally tractable without an additional approximation, known as Pólya-Gamma augmentation of the multinomial distribution. The full details of the augmentation procedure are described below in Appendix B. Here we will briefly sketch the main steps and describe the high level, augmented update equations. The Pólya-Gamma augmentation introduces datapoint-specific auxiliary variables $(\boldsymbol{\omega}_1^n, \boldsymbol{\omega}_0^n)$, that help us transform the log-probability of the multinomial distribution into a quadratic function [Polson et al., 2013, Linderman et al., 2015] over coefficients $(\boldsymbol{\beta}_1, \boldsymbol{\beta}_0)$, and latents $\boldsymbol{x}_1^n$. This quadratic form enables tractable update of $q\left(\boldsymbol{x}_1^n | z_1^n\right)$ in the form of a multivariate normal distribution, and a tractable updating of posteriors over coefficients $q\left(\boldsymbol{\beta}_1\right)$ and $q\left(\boldsymbol{\beta}_0\right)$.

With the introduction of the auxiliary variables the variational expectation and maximisation steps are expressed as

Update latents ('E-step')

$$q_t\left(\boldsymbol{x}_1^n | z_1^n\right) \propto \exp\left\{\mathbb{E}_{q_{t-1}(\boldsymbol{\beta}_1)q_{t-1}(\boldsymbol{\lambda}_1)}\left[\left\langle l\left(y^n, \boldsymbol{x}_1^n, \boldsymbol{\omega}_1^n, \boldsymbol{\beta}_1\right)\right\rangle_{q_{t-1}\left(\boldsymbol{\omega}_1^n | y^n\right)} + \ln p_{\boldsymbol{\lambda}_1}\left(\boldsymbol{x}_1^n | \boldsymbol{x}_0^n, z_1^n\right)\right]\right\}$$

$$q_t\left(\boldsymbol{\omega}_1 | y^n\right) \propto p\left(\boldsymbol{\omega}_1^n | y_n\right) \exp\left\{\mathbb{E}_{q_{t-1}(\boldsymbol{\beta}_1)q_t\left(\boldsymbol{x}_1^n | z_1^n\right)}\left[l\left(y^n, \boldsymbol{x}_1^n, \boldsymbol{\omega}_1^n, \boldsymbol{\beta}_1\right)\right]\right\}$$

$$q_t\left(\boldsymbol{\omega}_0 | z_1^n\right) \propto p\left(\boldsymbol{\omega}_0^n | z_1^n\right) \exp\left\{\mathbb{E}_{q_{t-1}(\boldsymbol{\beta}_0)}\left[l\left(z_1^n, \boldsymbol{x}_0^n, \boldsymbol{\omega}_0^n, \boldsymbol{\beta}_0\right)\right]\right\}$$

$$q_t\left(z_1^n\right) \propto \exp\left\{\mathbb{E}_{q_{t-1}(\boldsymbol{\Theta})}\left[\bar{l}_{z_1^n, t}\left(y^n, \boldsymbol{\beta}_1\right) + R_{z_1^n, t}\left(\boldsymbol{x}_0^n, \boldsymbol{\lambda}_1\right) + \bar{l}_t\left(z_1^n, \boldsymbol{x}_0^n, \boldsymbol{\beta}_0\right)\right]\right\}$$

Update parameters ('M-step')

$$q_t\left(\boldsymbol{\beta}_1\right) \propto \exp\left\{\sum_{n=1}^N \mathbb{E}_{q_t\left(\boldsymbol{x}_1^n, z_1^n\right)q_t\left(\boldsymbol{\omega}_1^n | y^n\right)}\left[l(y^n, \boldsymbol{x}_1^n, \boldsymbol{\beta}_1, \boldsymbol{\omega}_1^n)\right]\right\}$$

$$q_t\left(\boldsymbol{\beta}_0\right) \propto \exp\left\{\sum_{n=1}^N \mathbb{E}_{q_t\left(z_1^n\right)q_t\left(\boldsymbol{\omega}_0^n | z_1^n\right)}\left[l(z_1^n, \boldsymbol{x}_0^n, \boldsymbol{\beta}_0, \boldsymbol{\omega}_0^n)\right]\right\}$$

$$(12)$$

where we skipped the terms whose form did not change. $R_{z_1^n, t}\left(\boldsymbol{x}_0^n, \boldsymbol{\lambda}_1\right)$ reflects a contribution to $q(z_1^n)$ that depends on the expected log partition of the linear (Matrix Normal Gamma) likelihood $p_{\boldsymbol{\lambda}_1}(\boldsymbol{x}_1^n | \boldsymbol{x}_0^n, z_1^n)$. Note that the updates to each subset of posteriors (latents or parameters) have an analytic form due to the conditional conjugacy of the model. Importantly, both priors and posterior of the auxiliary variables are Pólya-Gamma distributed [Polson et al., 2013].

Finally, in the above update equations, we have replaced instances of the multinomial distribution $p\left(z | \boldsymbol{x}, \boldsymbol{\beta}\right)$ with its augmented form $p\left(\omega | z\right)e^{l(z, \boldsymbol{x}, \boldsymbol{\omega}, \boldsymbol{\beta})}$ where the function $l\left(\cdot\right)$ is quadratic with respect to the coefficients $\boldsymbol{\beta}$ and the input variables $\boldsymbol{x}$, leading to tractable update equations.

# B   Variational Bayesian Multinomial Logistic Regression

In this section, we focus on a single multinomial logistic regression model (not in the context of the CMN), but the ensuing variational update scheme derived in Appendix B.4 is applied in practice to both the gating network's parameters $\boldsymbol{\beta}_0$ as well as those of the final output likelihood for the class label $\boldsymbol{\beta}_1$.

## B.1   Stick-breaking reparameterization of a multinomial distribution

Multinomial logistic regression considers the probability that an outcome variable $y$ belongs to one of $K$ mutually-exclusive classes or categories. The probability of $y$ belonging to the $k^{\text{th}}$ class is given by the categorical likelihood:

$$p(y = k|\boldsymbol{x}, \boldsymbol{\beta}) = p_k \tag{13}$$

The problem of multinomial logistic regression is to identify or estimate the values of regression coefficients $\boldsymbol{\beta}$ that explain the relationship between some dataset of given continuous input regressors $\boldsymbol{X} = (\boldsymbol{x}^1, \boldsymbol{x}^2, \ldots, \boldsymbol{x}^N)$ and corresponding categorical labels $Y = (y^1, y^2, \ldots, y^N), y^n \in 1, 2, \ldots, K$.

We can use a stick-breaking construction to parameterize the likelihood over $y$ using a set of $K - 1$ stick-breaking coefficients: $\boldsymbol{\pi} = (\pi_1, \ldots, \pi_{K-1})$. Each coefficient is parameterized with an input regressor $\boldsymbol{x}$, and a corresponding set of regression weights $\boldsymbol{\beta}_j$. Stick-breaking coefficient $\pi_j$ is then given by a sigmoid transform of the product of the regression weights and the input regressors:

$$\pi_j = \sigma\left(\boldsymbol{\beta}_j\left[\boldsymbol{x}; 1\right]\right) ,$$
$$\text{where } \sigma\left(\boldsymbol{\beta}_j\left[\boldsymbol{x}; 1\right]\right) = \frac{1}{1 + \exp\left\{-\boldsymbol{\beta}_j\left[\boldsymbol{x}; 1\right]\right\}} ,$$
$$\text{and } \boldsymbol{\beta}_j\left[\boldsymbol{x}; 1\right] = \sum_{i=1}^{d} w_{j,i} x_i + a_j. \tag{14}$$

The outcome likelihood is then obtained via stick breaking transform[5] as follows

$$p_k = \pi_K \prod_{j=1}^{K-1} (1 - \pi_j) = \sigma\left(\boldsymbol{\beta}_K\left[\boldsymbol{x}; 1\right]\right) \prod_{j=1}^{K-1} \left(1 - \sigma\left(\boldsymbol{\beta}_j\left[\boldsymbol{x}; 1\right]\right)\right) = \prod_{j=1}^{K-1} \frac{\exp\left\{\boldsymbol{\beta}_j\left[\boldsymbol{x}; 1\right]\right\}}{1 + \exp\left\{\boldsymbol{\beta}_j\left[\boldsymbol{x}; 1\right]\right\}} \tag{15}$$

where $\pi_K = 1$, and $\boldsymbol{\beta}_K = \vec{0}$.

Finally, we can express the likelihood in the form of a Categorical distribution as

$$\text{Cat}\left(y; \boldsymbol{x}, \boldsymbol{\beta}\right) = \prod_{k=1}^{K-1} \frac{\left(\exp\left\{\boldsymbol{\beta}_k\left[\boldsymbol{x}; 1\right]\right\}\right)^{\delta_{k,y}}}{\left(1 + \exp\left\{\boldsymbol{\beta}_k\left[\boldsymbol{x}; 1\right]\right\}\right)^{N_{k,y}}} . \tag{16}$$

where $N_{k,y} = 1$ for $k \leq y$, and $N_{k,y} = 0$ otherwise (or $N_{k,y} = 1 - \sum_{j=1}^{k-1} \delta_{j,y}$), and $\delta_{k,y} = 1$ for $k = y$ and is zero otherwise.

### B.2 Pólya-Gamma augmentation

The *Pólya-Gamma augmentation* scheme [Polson et al., 2013, Linderman et al., 2015, Durante and Rigon, 2019] is defined as

$$\frac{\left(e^\psi\right)^a}{\left(1 + e^\psi\right)^b} = 2^{-b} e^{\kappa\psi} \int_0^\infty e^{-\omega\psi^2/2} p(\omega) \, \mathrm{d}\omega \tag{17}$$

where $\kappa = a - b/2$ and $p\left(\omega|b, 0\right)$ is the density of the Pólya-Gamma distribution $\text{PG}(b, 0)$ which does not depend on $\psi$. The useful properties of the Pólya-Gamma are the exponential tilting property expressed as

$$PG(\omega; b, \psi) = \frac{e^{-\omega\psi^2/2} PG(\omega; b, 0)}{\mathbb{E}\left[e^{-\omega\psi^2/2}\right]} \tag{18}$$

---

[5]This blog post has helpful discussion on the stick-breaking form of the multinomial logistic likelihood and provides more intuition behind its functional form.

the expected value of $\omega$, and $e^{-\omega\psi^2/2}$ given as

$$\mathbb{E}\left[\omega\right] = \int_o^\infty \omega PG(\omega; b, \psi)\, \mathrm{d}\omega = \frac{b}{2\psi} \tanh\left(\frac{\psi}{2}\right),$$

$$\mathbb{E}\left[e^{-\omega\psi^2/2}\right] = \cosh^{-b}\left(\frac{\psi}{2}\right) \tag{19}$$

and the Kulback-Leibler divergence between $q\left(\omega\right) = PG(\omega; b, \psi)$ and $p\left(\omega\right) = PG(\omega; b, 0)$ obtained as

$$D_{KL}\left[q\left(\omega\right)||p\left(\omega\right)\right] = -\mathbb{E}\left[\omega\right]\frac{\psi^2}{2} + b\ln\cosh\left(\frac{\psi}{2}\right) = -\frac{b\psi}{4}\tanh\left(\frac{\psi}{2}\right) + b\ln\cosh\left(\frac{\psi}{2}\right). \tag{20}$$

We can express the likelihood function in Equation (16) using the augmentation as

$$p(y,\boldsymbol{\omega}|\boldsymbol{\psi}) = p\left(y|\boldsymbol{\psi}\right)p\left(\boldsymbol{\omega}|y,\boldsymbol{\psi}\right) = \prod_{k=1}^{K-1} 2^{-b_{k,y}} e^{\kappa_{k,y}\psi_k - \omega_k\psi_k^2/2}\mathrm{PG}(\omega_k; b_{k,y}, 0)$$

$$p\left(y|\boldsymbol{\psi}\right) = \prod_{k=1}^{K-1} 2^{-b_{k,y}} e^{\kappa_{k,y}\psi_k} \int_0^\infty e^{-\omega_k\psi_k^2/2}\mathrm{PG}(\omega_k; b_{k,y}, 0)\,\mathrm{d}\omega_k \tag{21}$$

$$p\left(\boldsymbol{\omega}|y,\boldsymbol{\psi}\right) = \prod_{k=1}^{K-1} \mathrm{PG}\left(\omega_k; b_{k,y}, \psi_k\right)$$

where $b_{k,y} \equiv N_{k,y}$, $\kappa_{k,y} = \delta_{k,y} - N_{k,y}/2$, and $\psi_k = \boldsymbol{\beta_k}\left[\boldsymbol{x}; 1\right]$. Given a prior distribution $p\left(\boldsymbol{\psi}\right) = p\left(\boldsymbol{\beta}\right)p\left(\boldsymbol{x}\right)$, we can write the joint $p\left(y,\boldsymbol{\omega},\boldsymbol{\psi}\right)$ as

$$p\left(y,\boldsymbol{\omega},\boldsymbol{\psi}\right) = p\left(\boldsymbol{\omega}|y\right)p\left(\boldsymbol{\psi}\right)e^{l(y,\boldsymbol{\psi},\boldsymbol{\omega})},$$

$$l\left(y,\boldsymbol{\psi},\boldsymbol{\omega}\right) = \sum_{k=1}^{K-1} l_k\left(y,\psi_k,\omega_k\right), \tag{22}$$

$$l_k\left(y,\psi_k,\omega_k\right) = \kappa_{y,k}\psi_k - b_{y,k}\ln 2 - \omega_k\psi_k^2/2.$$

## B.3  Evidence lower-bound

Given a set of observations $\boldsymbol{\mathcal{D}} = \left(y^1,\ldots,y^N\right)$ the augmented joint distribution can be expressed as

$$p\left(\boldsymbol{\mathcal{D}},\boldsymbol{\Omega},\boldsymbol{X},\boldsymbol{\beta}\right) = p\left(\boldsymbol{\beta}\right)\prod_{n=1}^{N} p\left(\boldsymbol{x}^n\right)p\left(\boldsymbol{\omega}^n|y^n\right)e^{l(y^n,\boldsymbol{\psi}^n,\boldsymbol{\omega}^n)}$$

We can express the evidence lower-bound (ELBO) as

$$\mathcal{L}(q) = E_{q(\boldsymbol{\Omega})q(\boldsymbol{X})q(\boldsymbol{\beta})}\left[-\ln q\left(\boldsymbol{\beta}\right) + \sum_{n=1}^{N}\ln\frac{p\left(y^n,\boldsymbol{\psi}^n,\boldsymbol{\omega}^n\right)}{q\left(\boldsymbol{\omega}^n\right)q\left(\boldsymbol{x}^n\right)}\right]$$

$$= E_{q(\boldsymbol{\Omega})q(\boldsymbol{X})q(\boldsymbol{\beta})}\left[\ln\frac{p\left(\boldsymbol{\beta}\right)}{q\left(\boldsymbol{\beta}\right)} + \sum_{n=1}^{N}l\left(y^n,\boldsymbol{\psi}^n,\boldsymbol{\omega}^n\right) + \ln\frac{p\left(\boldsymbol{\omega}^n|y^n\right)}{q\left(\boldsymbol{\omega}^n\right)} + \ln\frac{p\left(\boldsymbol{x}^n\right)}{q\left(\boldsymbol{x}^n\right)}\right] \tag{23}$$

$$\geq \ln p\left(\boldsymbol{\mathcal{D}}\right)$$

where we use the following forms for the approximate posterior

$$q\left(\boldsymbol{\Omega}|Y\right) = \prod_{n=1}^{N} q\left(\boldsymbol{\omega}^n|y^n\right) = \prod_{n=1}^{N}\prod_{k=1}^{K-1} PG\left(b_{k,y^n}, \xi_{k,n}\right) \ ,$$

$$q\left(\boldsymbol{X}\right) = \prod_{n=1}^{N} q\left(\boldsymbol{x}^n\right) = \prod_{n=1}^{N} \mathcal{N}\left(\boldsymbol{x}^n; \boldsymbol{\mu}^n, \boldsymbol{\Sigma}^n\right) \ , \tag{24}$$

$$q\left(\boldsymbol{\beta}\right) = \prod_{k=1}^{K-1} \mathcal{N}\left(\boldsymbol{\beta}_k; \boldsymbol{\mu}_k, \boldsymbol{\Sigma}_k\right) \ .$$

## B.4 Coordinate ascent variational inference for the PG-augmented model

The mean-field assumption in Equation (24) allows the implementation of a simple CAVI algorithm [Wainwright et al., 2008, Beal, 2003, Hoffman et al., 2013, Blei et al., 2017] which sequentially maximizes the evidence lower bound in Equation (23) with respect to each factor in $q\left(\boldsymbol{\Omega}|Y\right) q\left(\boldsymbol{X}\right) q\left(\boldsymbol{\beta}\right)$, via the following updates:

Update to latents ('E-step')

$$q^{(t,l)}\left(\boldsymbol{x}^n\right) \propto p\left(\boldsymbol{x}^n\right) \exp\left\{\mathbb{E}_{q^{(t-1)}(\boldsymbol{\beta})q^{(t,l-1)}(\boldsymbol{\omega}^n)}\left[l\left(y^n, \boldsymbol{\psi}^n, \boldsymbol{\omega}^n\right)\right]\right\}$$

$$q^{(t,l)}\left(\omega_k^n|y^n\right) \propto p\left(\omega_k^n|y^n\right) \exp\left\{\mathbb{E}_{q^{(t-1)}(\boldsymbol{\beta})q^{(t,l)}(\boldsymbol{x}^n)}\left[l_k\left(y^n, \psi_k^n, \omega_k^n\right)\right]\right\}$$

$$\forall n \in \{1, \ldots, N\}, \text{ and for } q^{(t,0)}\left(\boldsymbol{\omega}^n|y^n\right) = q^{(t-1,L)}\left(\boldsymbol{\omega}^n|y^n\right) \tag{25}$$

Update to parameters ('M-step')

$$q^{(t)}\left(\boldsymbol{\beta}_k\right) \propto \exp\left\{\sum_{n=1}^{N}\mathbb{E}_{q^{(t)}(\boldsymbol{x}^n)q^{(t)}(\boldsymbol{\omega}^n|y^n)}\left[l\left(y^n, \boldsymbol{\psi}^n, \boldsymbol{\omega}^n\right)\right]\right\}$$

at each iteration $t$, and multiple local iteration $l$ during the variational expectation step—until the convergence of the ELBO.

Specifically, the update equations for the parameters of the latents (the 'E-step') are:

$$q^{(t,l)}\left(\boldsymbol{x}^n\right) \propto \mathcal{N}\left(\boldsymbol{x}^n; 0, -2\boldsymbol{\lambda}_{2,0}\right) \exp\left\{\sum_{k=1}^{K}\kappa_{k,y^n}\text{Tr}\left(\boldsymbol{\mu}_k^{(t-1)}\left[\boldsymbol{x}^n; 1\right]^T\right) - \frac{\langle\omega_k\rangle}{2}\text{Tr}\left(\boldsymbol{M}_k^{(t-1)}\left[\boldsymbol{x}^n; 1\right]\left[\boldsymbol{x}^n; 1\right]^T\right)\right\}$$

$$\boldsymbol{\lambda}_1^{(n,t,l)} = \sum_{k=1}^{K-1}\left\{\kappa_{k,y^n}\left[\boldsymbol{\mu}_k^{(t-1)}\right]_{1:D} - \langle\omega_k^n\rangle_{t,l-1}\left[\boldsymbol{M}_k^{(t-1)}\right]_{D+1,1:D}\right\}$$

$$\boldsymbol{\lambda}_2^{(n,t,l)} = \boldsymbol{\lambda}_{2,0} - \frac{1}{2}\sum_{k=1}^{K-1}\langle\omega_k^n\rangle_{t,l-1}\left[\boldsymbol{M}_k\right]_{1:D,1:D}$$

$$\boldsymbol{M}_k^{(t-1)} = \boldsymbol{\Sigma}_k^{(t-1)} + \boldsymbol{\mu}_k^{(t-1)}\left[\boldsymbol{\mu}_k^{(t-1)}\right]^T \tag{26}$$

and

$$q^{(t,l)}\left(\omega_k^n|y^n\right) \propto e^{-\omega_k^n\langle\psi_k^2\rangle/2}PG\left(\omega_k^n; b_{k,y^n}, 0\right)$$

$$\xi_k^n = \sqrt{\mathbb{E}_{q^{(t-1)}(\boldsymbol{\beta})q^{(t,l)}(\boldsymbol{x}^n)}\left[\psi_k^2\right]}$$

$$\xi_k^n = \sqrt{\text{Tr}\left(\boldsymbol{M}_k^{(t-1)}\hat{\boldsymbol{M}}^{(n,t,l)}\right)} \tag{27}$$

where $\hat{\boldsymbol{M}}^{(n,t,l)} = \begin{pmatrix} \boldsymbol{M}^{(n,t,l)} & \boldsymbol{\mu}^{(n,t,l)} \\ \left[\boldsymbol{\mu}^{(n,t,l)}\right]^T & 1 \end{pmatrix}$, and $\boldsymbol{M}^{(n,t,l)} = \boldsymbol{\Sigma}^{(n,t,l)} + \boldsymbol{\mu}^{(n,t,l)}\left[\boldsymbol{\mu}^{(n,t,l)}\right]^T$.

Similarly, for the parameter updates ('M-step') we get

$$q^{(t)}\left(\boldsymbol{\beta}_k\right) \propto \mathcal{N}\left(\boldsymbol{\beta}_k; 0, -2\boldsymbol{\lambda}'_{2,0}\right) \exp\left\{\sum_{n=1}^{N} \kappa_{k,y^n} \operatorname{Tr}\left(\hat{\boldsymbol{\mu}}^{(n,t)}\boldsymbol{\beta}_k^T\right) - \frac{\langle\omega_k\rangle_t^n}{2}\operatorname{Tr}\left(\hat{\boldsymbol{M}}_i^{(t)}\boldsymbol{\beta}_k\boldsymbol{\beta}_k^T\right)\right\}$$

$$\boldsymbol{\lambda}_{k,1}^{(t)} = \sum_i \kappa_{k,y^n}\hat{\boldsymbol{\mu}}^{(n,t)} \tag{28}$$

$$\boldsymbol{\lambda}_{k,2}^{(t)} = \boldsymbol{\lambda}'_{2,0} - \frac{1}{4}\sum_{n=1}^{N} \frac{b_{k,y^n}}{\xi_k^{(n,t)}}\tanh\left(\frac{\xi_k^{(n,t)}}{2}\right)\hat{\boldsymbol{M}}^{(n,t)}$$

where $\hat{\boldsymbol{\mu}}^{(n,t)} = \left[\boldsymbol{\mu}^{(n,t)}; 1\right]$.

## C  Hyperparameters

### C.1  Common hyperparameters

For the Bayesian methods (CAVI, NUTS, and BBVI), we used the same form for the CMN priors (see Equation (6) for their parameterization) and fixed the prior parameters to the following values, used for all datasets: $v_0 = 10$, $a_0 = 2$, $b_0 = 1$, $\sigma_0, \sigma_1 = 5$. For all datasets, we fixed the dimension of the continuous latent $\boldsymbol{x}_1$ to be $h = L - 1$, where $L$ is the number of classes. For the Pinwheels dataset (see Appendix D.1 below), we set the number of linear experts (and hence the dimension of the discrete latent $\boldsymbol{z}_1$) at $K = 10$, while for all other datasets we used $K = 20$.

### C.2  Posterior Initialization for CAVI-CMN

We initialized the posterior parameters of the Matrix Normal Gamma distributions for each linear expert in the conditional mixture layer of the network in the following way:

- Each element $M_{ij}$ of the posterior mean matrix $\boldsymbol{M}_{k\in 1:K}$ was independently sampled from a uniform distribution $\mathcal{U}(\frac{-3}{\sqrt{d}}, \frac{3}{\sqrt{d}})$, where $\mathcal{U}(lb, ub)$ represents the uniform distribution with bounds $lb$ and $ub$, and $d$ is the input dimension.

- The initial posterior values of $\boldsymbol{V}_{k\in 1:K}$ were set to identity matrices.

- The initial posteriors for $a_{k\in 1:K}$ and $b_{k\in 1:K}$ were set to 2.0 and 1.0, respectively.

We initialized the posterior parameters of the two Multinomial Logistic Regressions (one for the conditional mixture layer, one for the terminal layer) in the following way:

- The posterior mean and covariance of the $k^{\text{th}}$ row of stick-breaking weights were multivariate normal distributions $\mathcal{N}(\boldsymbol{\beta}_k; \boldsymbol{\mu}_k, \boldsymbol{\Sigma}_k)$ with the following parameters $\boldsymbol{\mu}_k = \left(0, 0, \ldots, -\log(K-k)\right)$, $\Sigma_k = 0.01\,I_D$. The mean of the $k^{\text{th}}$ bias term was initialized to a 'stick-breaking correction term' $-\log(K-k)$ in order to induce a flat prior over the $K$ categories. In the absence of this correction, the stick-breaking parameterization assigns non-uniform probability across categories, whereby categories with lower indices in the ordering $k \in (1, 2, \ldots, K)$ are assigned higher likelihood *a priori*.

### C.3  Maximum Likelihood Estimation

For gradient-based optimization of the loss function (the negative log likelihood), we used the AdaBelief optimizer with parameters set to its default values as introduced in Zhuang et al. [2020] ($\alpha = 1e-3$, $\beta_1 = 0.9$, $\beta_2 = 0.999$), and ran the optimization for $20,000$ steps. This implements deterministic gradient descent, not stochastic gradient descent, because we fit the model in 'full-batch' mode, i.e., without splitting the data into mini-batches and updating model parameters using noisy gradient estimates.

## C.4   No U-Turn Sampler

Markov Chain Monte Carlo converges in distribution to samples from a target distribution, so for this method we obtain samples from a joint distribution $p(\boldsymbol{A}_{1:K}, \boldsymbol{\Sigma}_{1:K}^{-1}, \boldsymbol{\beta}_0, \boldsymbol{\beta}_1 | Y, \boldsymbol{X}_0)$ that approximate the true posterior. We used 800 warm-up steps, 16 independent chains, and 64 samples for each chain.

## C.5   Black box variational inference

While BBVI does not require conjugate relationships in the generative model, we use the same CMN model and variational distributions as we use for CAVI-CMN, in order to ensure fair comparison. For stochastic optimization, we used the AdaBelief optimizer with learning rate $\alpha = 5e-3$ $\beta_1 = 0.9$, $\beta_2 = 0.999$, used 8 samples to estimate the ELBO gradient (the `num_particles` argument of the `Trace_ELBO()` class), and ran the optimizer for $20,000$ steps).

# D   Dataset Descriptions

We fit all inference methods using different training set sizes, where each next training set was twice as large as the previous. This was done in order to study the robustness of performance in the low data regime. For each training size, we used the same test-set to evaluate performance. The test set was ensured to have the same relative class frequencies as in the training set(s). For each inference method and examples set size, we fit using the same batch of training data, but with 16 randomly-initialized models (different initial posterior samples or parameters).

## D.1   Pinwheels Dataset

The pinwheels dataset is a synthetic dataset designed to test a model's ability to handle nonlinear decision boundaries and data with non-Gaussian densities. The dataset consists of multiple clusters arranged in a pinwheel pattern, posing a challenging task for mixture models [Johnson et al., 2016] due to the curved and elongated spatial distributions of the data. The structure of the pinwheels dataset is determined by 4 parameters: the number of clusters or distinct spirals; the angular deviation, which defines how far the spiralling clusters deviate from the origin; the tangential deviation, which defines the noise variance of 2-D points within each cluster; and the angular rate, which determines the curvature of each spiral. For evaluating the four methods (CAVI-CMN, MLE, BBVI, and NUTS) on the synthetic pinwheels dataset, we generated a dataset with 5 clusters, with an angular deviation of 0.7, tangential deviation of 0.3 and angular rate of 0.2. We selected these values by looking at the maximum achieved test accuracy across all the methods for different parameter combinations and tried to upper-bound it 80%, which provides a low enough signal-to-noise ratio to be able to meaningfully show differences in probabilistic metrics like calibration and WAIC. For pinwheels, we trained using train sizes 50 to 1600, doubling the number of training examples at each successive training set size. We tested using 500 held-out test examples generated using the same parameters as used for the training set(s).

## D.2   Waveform Domains Dataset

The Waveform Domains dataset consists of synthetic data generated to classify three different waveform patterns, where each class is described by 21 continuous attributes [Breiman and Stone, 1988]. For waveform domains, we fit each model on train sizes ranging from 60 to 3840 examples, and tested on a held-out size of 1160 examples. See here for more information about the dataset.

## D.3   Vehicle Silhouettes Dataset

This dataset involves classifying vehicle silhouettes into one of four types (bus, van, or two car models) based on features extracted from 2D images captured at various angles [Mowforth and Shepherd]. We fit each model on train sizes ranging from 20 to 650 examples, and tested on a held-out size of 205 examples. See here for more information about the dataset.

### D.4 Rice Dataset

The Rice dataset contains measurements related to the classification of rice varieties, specifically Cammeo and Osmancik [mis, 2019]. We fit each model on train sizes ranging from 40 to 2560 examples, and tested on a held-out size of 1250. See here for more information about the dataset.

### D.5 Breast Cancer Dataset

The 'Breast Cancer Diagnosis' dataset [Wolberg et al., 1995] contains features extracted from breast mass images, which are then used to classify tumors as malignant or benign. See here for more information about the dataset. We fit each model on train sizes ranging from 25 to 400 examples, and tested on a held-out size of 169.

### D.6 Sonar (Mines vs Rocks) Dataset

The Sonar (Mines vs Rocks) dataset consists of sonar signals bounced off metal cylinders and rocks under various conditions. The dataset includes 111 patterns from metal cylinders (mines) and 97 patterns from rocks. Each pattern is represented by 60 continuous attributes corresponding to the energy within specific frequency bands [Sejnowski and Gorman]. The task is to classify each pattern as either a mine (M) or a rock (R). For this dataset, we fit each model on train sizes ranging from 8 to 128 examples and tested on a held-out size of 80 examples. See here for more information about the dataset.

### D.7 Banknote Authentication Dataset

The 'Banknote Authentication' dataset [Lohweg, 2013] contains features extracted from images of genuine and forged banknotes. It is primarily used for binary classification tasks to distinguish between authentic and counterfeit banknotes. See here for more information about the dataset.

## E UCI Performance Results

In Figures 3 to 8 we report the same performance and runtime metrics as in Figure 2 for 7 UCI datasets, and find that with the exception of the Sonar dataset, CAVI performs competitively with or better than MLE on all datasets, and always outperforms MLE in terms of LPD and ECE. Runtime scaling is similar as reported for the Pinwheels dataset in the main text; CAVI-CMN always converges in fewer steps and is faster than BBVI, and either outperforms or is competitive with MLE in terms of runtime.

## F Relative runtime results

### F.1 Runtime comparison

In this subsection, we analyze the runtime efficiency of the MLE and BBVI algorithms for CMN models, in comparison to a CAVI-based approach. The focus is on comparing the computation time as the number of parameters increases along different components of the model.

To ensure comprehensive comparison, we varied the complexity of the models by adjusting the number of components, the dimensionality of the input space, and the dimensionality of the continuous latent variables. These modifications effectively increase the number of parameters allowing us to observe how each algorithm scales with different ways of manipulating of model complexity.

In Figure 9 we plot the relative runtimes of Maximum Likelihood, CAVI, and BBVI (proportional to the runtime of the least complex variant), as we increase the number of parameters along different dimensions. This shows how CAVI-CMN scales competitively with gradient-based methods like BBVI and Maximum Likelihood Estimation. However, the rightmost subplot indicates that as we increase the dimensionality of the latent variable $X_1$, CAVI-CMN scales more dramatically than the other two methods. This inherits from the computational overhead of matrix operations required by storing multivariate Gaussians posteriors over each continuous latent, i.e.,

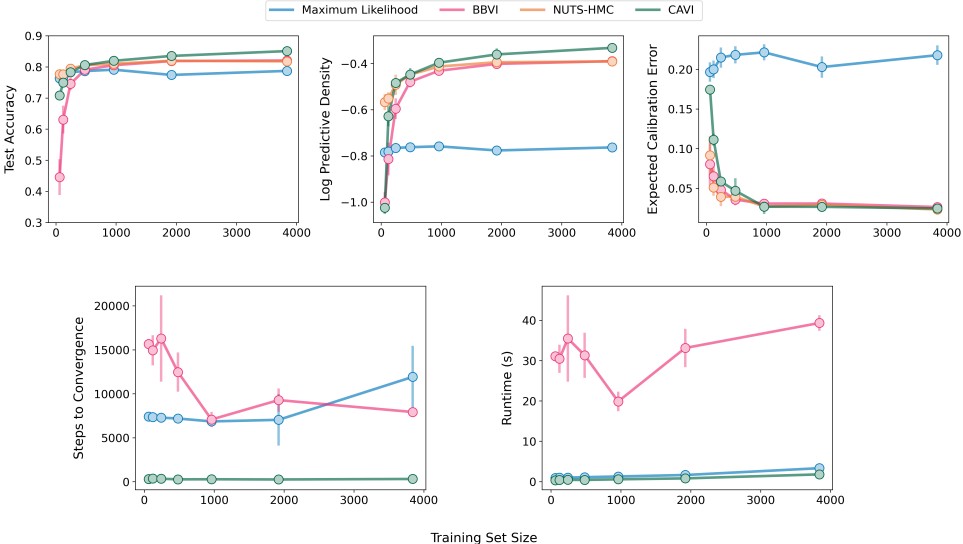

Figure 3: Performance and runtime results of the different models on the 'Waveform Domains' dataset. Descriptions of each subplot are same as in the Figure 2 legend.

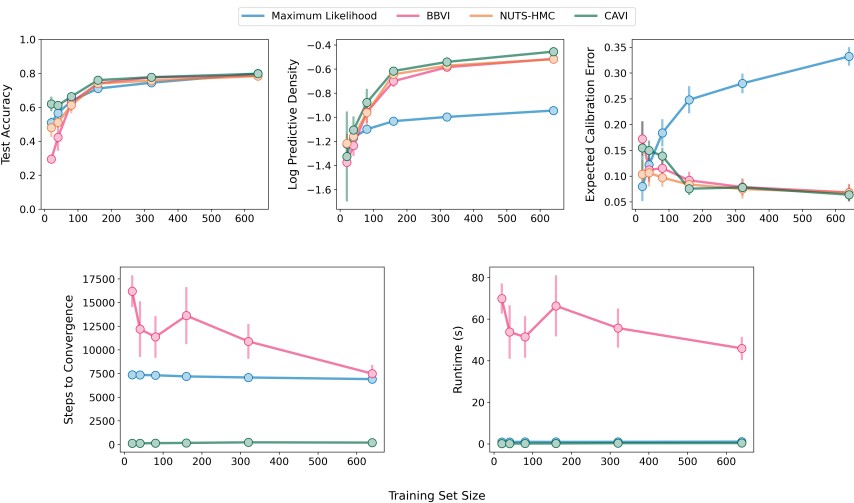

Figure 4: Performance and runtime results of the different models on the 'Vehicle Silhouettes' dataset. Descriptions of each subplot are same as in the Figure 2 legend.

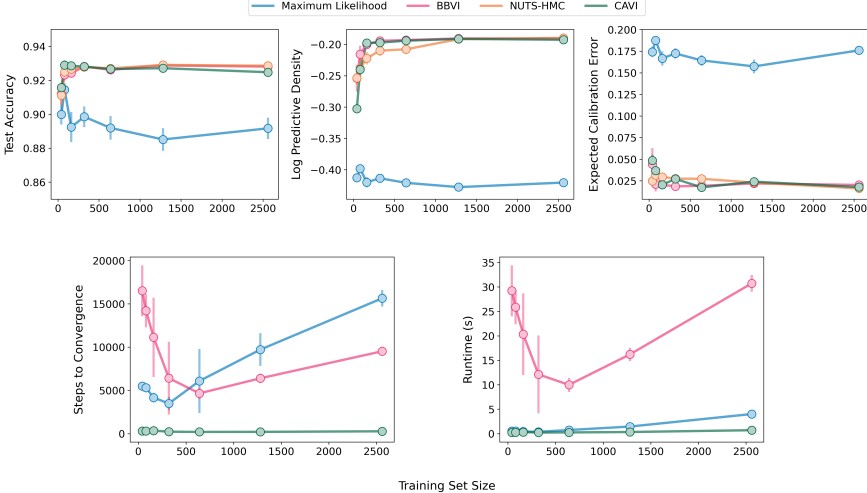

Figure 5: Performance and runtime results of the different models on the 'Rice' dataset. Descriptions of each subplot are same as in the Figure 2 legend.

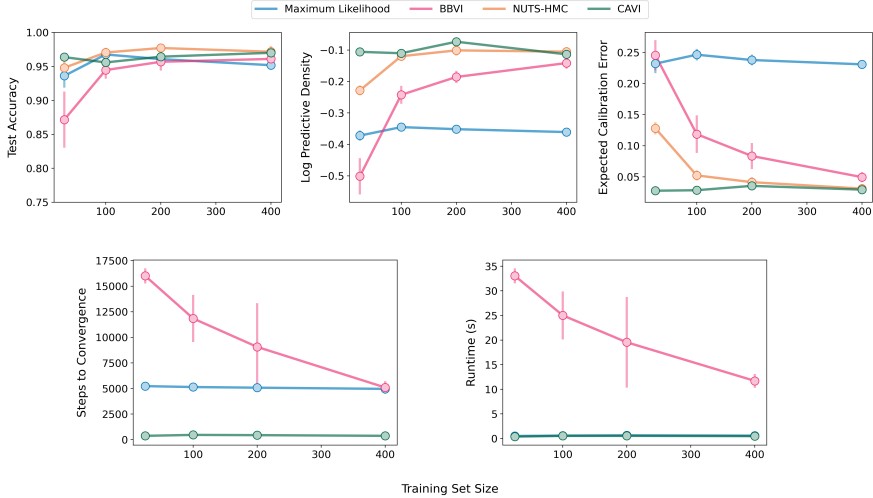

Figure 6: Performance and runtime results of the different models on the 'Breast Cancer' dataset. Descriptions of each subplot are same as in the Figure 2 legend.

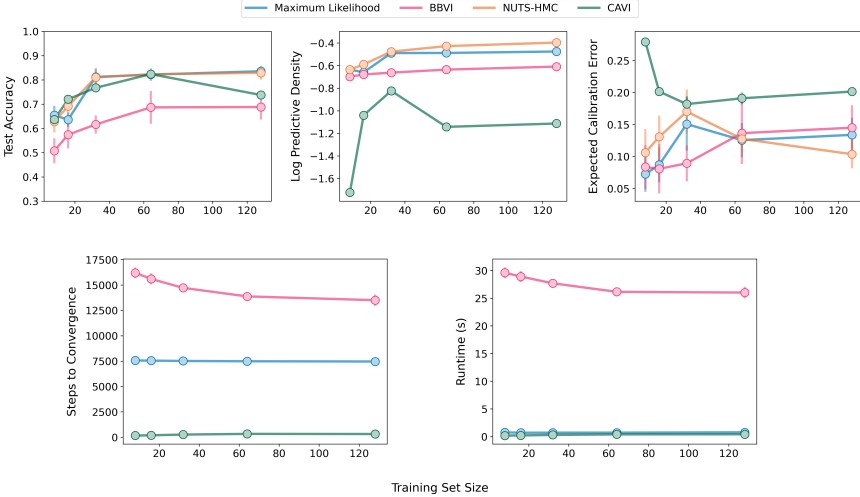

Figure 7: Performance and runtime results of the different models on the 'Connectionist Bench (Sonar, Mines vs. Rocks)' dataset. Descriptions of each subplot are same as in the Figure 2 legend..

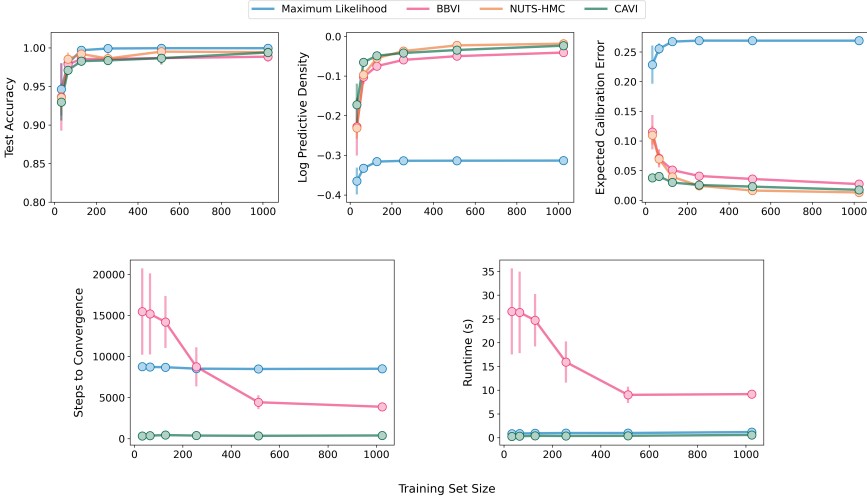

Figure 8: Performance and runtime results of the different models on the 'Banknote Authentication' dataset. Descriptions of each subplot are same as in the Figure 2 legend.

$q(\boldsymbol{x}_1^n|z_1^n) = \mathcal{N}(\boldsymbol{x}_1^n; \boldsymbol{\mu}_1^n, \boldsymbol{\Sigma}_1^n)$. Running the CAVI algorithm involves operations (like matrix inversions and matrix-vector products) whose (naive) complexity is quadratic in matrix size. This explains the nonlinear scaling of runtime as a function of $h$, the dimension of $\boldsymbol{X}_1$. There are several ways to address this issue:

- **Low-Rank Approximations**: Use low-rank approximations to the covariance matrix $\boldsymbol{\Sigma}_1^n$ (e.g., Cholesky or eigendecompositions).

- **Diagonal Covariance Structure**: Further constrain the covariance structure of $q(\boldsymbol{x}_1^n)$ by forcing the latent dimensions to be independent in the posterior, i.e., $q(\boldsymbol{x}_1^n|z_1^n) = \prod_{i=1}^h \mathcal{N}(x_{i,1}^n; \mu_{i,1}^n, (\sigma_{i,1}^n)^2)$. This would then mean that the number of parameters to store would only grow as $K(2h)$ in the size of the dataset, rather than as $K(h + O(h^2))$.

- **Full Mean-Field Approximation**: Enforce a full mean-field approximation between $\boldsymbol{X}_1$ and $Z_1$, so that one only needs to store $q(\boldsymbol{x}_1^n)q(z_1^n)$ rather than $q(\boldsymbol{x}_1^n|z_1^n)q(z_1^n)$. This would reduce the number of multivariate normal parameters that would have to be stored and operated upon by a factor of $K$.

- **Shared Conditional Covariance Structure**: Assume that the conditional covariance structure is shared across all training data points, i.e., $\boldsymbol{\Sigma}_1^n = \boldsymbol{\Sigma}_1$, for all $n \in \{1, 2, \ldots, n\}$.

All of these adjustments would help mitigate the quadratic runtime scaling of CAVI-CMN as the latent dimension $h$ increases.

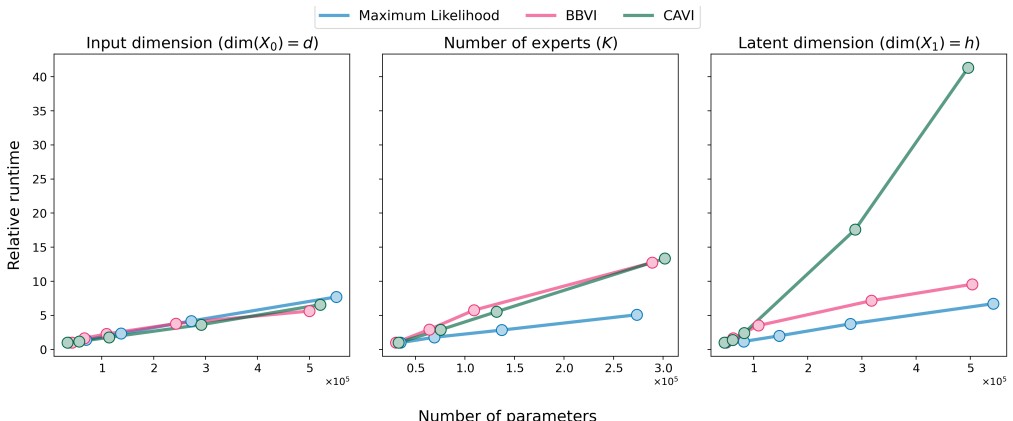

Figure 9: Relative scaling of fitting time in seconds for Maximum Likelihood, BBVI, and CAVI, as a function of the number of parameters. The number of parameters itself was manipulated in three illustrative ways: changing the input dimension $d$, changing the number of linear experts $K$ in the conditional mixture layer, and changing the dimensionality of the continuous latent variable $h$.

## G  Model Convergence Determination

For each inference algorithm, the number of iterations taken to converge was determined by running each algorithm for a sufficiently high number of gradient (respectively, CAVI update) steps such that the ELBO (or log likelihood - LL - for MLE) stopped significantly changing. This was determined (through anecdotal inspection over many different initializations and runs across the different UCI datasets) to be 20,000 gradient steps for BBVI, 20,000 gradient steps for MLE, and 500 combined CAVI update steps for CAVI-CMN. To determine the time taken to sufficiently converge, we recorded the value of the ELBO or LL at each iteration, and fit an exponential decay function to the negative of each curve. The parameters of the estimated exponential decay were then used to determine the time at which the curve decayed to 95% decay of its value. This time was reported as the number of steps taken to converge.

