# OpenReview forum: "Gradient-free variational learning with conditional mixture networks"
_NeurIPS.cc/2024/Workshop/BDU — NeurIPS BDU Workshop 2024 Poster_

### Official Review · Reviewer_3ver · 2024-09-24
**The proposed method is well-developed, the paper is well-presented, and it provides a solid empirical comparison to other standard methods.**

**Rating:** 7
**Confidence:** 2

**Review:**

The authors introduce CAVI-CMN, a gradient-free variational method for training conditional mixture networks. The paper's key contribution is that CAVI-CMN outperforms other variational methods, such as BBVI, in terms of speed, while offering comparable efficiency to MLE and maintaining strong predictive accuracy.

The presentation of the paper is quite good. The empirical results clearly show the advantage of using CAVI-CMN compared to the baseline method.

Some minor comments:
1. The empirical results are impressive, but I wonder about their scalability on larger models or datasets than those currently presented.
2. Although the PG augmentation scheme is explained in the appendix, it is not clearly described in the main paper. For instance, it is mentioned at the beginning of Section 2 and in Section 2.2, but a clearer one-sentence explanation might be helpful.
3. In Figure 1 and all other performance comparison figures, the green line labeled "CAVI" should be "CAVI-CMN," correct?
4  For advanced readers, it is clear why NUTS are not present in Figure 1, as it is expected to be the slowest among the methods. However, it might be better to include a clarification somewhere.
5. Unless I am missing something, I believe a more detailed description of the neural network used, such as the number of parameters, would help the reader better understand the scale of the model.

---

### Official Review · Reviewer_29cT · 2024-09-27
**BDU Review**

**Rating:** 8
**Confidence:** 3

**Review:**

# Overview

Thanks to the authors for the interesting and well-organized paper. The overall quality is good, with clear motivation and claims, as well as significant experimental results to back up those claims. The paper demonstrates a novel application of Coordinate Ascent Variational Inference (CAVI) to fitting fully (approximate) Bayesian models of a particular structure. My general impression is that CAVI has largely fallen out of favor compared to Black Box Variational Inference (BBVI), which allows for more general approximate posteriors. However, the authors show that, for this particular class of models, CAVI yields strong results that are competitive with BBVI (and NUTS) while requiring significantly less wall-clock time. The wall-clock time is in fact similar to MLE, while retaining a full posterior over parameters. The significance of the paper may be reduced by the fact the structure of the model is limited by conjugacy requirements (in order to yield efficient updates): the authors mention that the could be expanded to deeper architectures, but I'm not convinced that this would gain anything if each layer is linear? I suppose it depends on what "deeper architectures" means concretely. However, all research has limitations: The paper's contribution is still meaningful within context.

## Pros
- Clearly written motivation.
- Well-analyzed experimental results that show the strengths of CAVI-CMN compared to other parameter fitting/inference approaches.
  - Experiments include both synthetic and real datasets.
  - Experiments compare to relevant baselines.
  - Experiments illustrate scaling of accuracy and running time based on training set size (and scaling of running time, w.r.t. other parameters, in appendix).
  - Metrics for experiments are well-chosen to illustrate important aspects of classification (e.g., not just accuracy or LPD, but also model calibration) and show advantages of maintaining a full posterior over model parameters.

## Cons
- Section **2.1 The conditional mixture network** was not very clear.
- A more thorough analysis might also show the scaling of accuracy with respect to number of experts in the mixture, perhaps for some subset of problems.
- Although it may be a little out-of-scope, I found myself wondering how this particular model (i.e., a CMN, fit by whatever means) compares to a more general model, such as a fully Bayesian neural network with some reasonable architecture. In other words, the paper does a good job of arguing that CAVI-CMN produces the best CMN, is such a model particularly _good_ at classification, compared to other potentially more flexible models?

# Suggestions
## Major
- Section **2.1 The conditional mixture network** could be much clearer. It's not so much that the structure of the model itself is hard to grasp once you've digested it, but the presentation could use some work.
  - Instead of "a condtional mixture of linear experts, which outputs a joint continuous-discrete latent $\left( \mathbf{x}\_{1} \in \mathbb{R}^{h}, z_{1} \in \\{ 1, \ldots, K \\} \right)$", I would write "a condtional mixture of linear experts, **each of** which outputs a joint continuous-discrete latent $\left( \mathbf{x}\_{1} \in \mathbb{R}^{h}, z_{1} \in \\{ 1, \ldots, K \\} \right)$".
  - A conceptual diagram would help immensely.
  - Although I recognize that labeling can often be a matter of taste, I would re-label as follows:
    - $\mathbf{x}_{0} \rightarrow \mathbf{x}$
    - $\mathbf{x}_{1} \rightarrow \mathbf{z}\_{1}$
    - $z_{1} \rightarrow z\_{2}$

  and similarly for capital-letter collections. This would make the inputs $x$, outputs $y$, and all latents $z$. Even if you decide to keep the labeling as-is, I don't see any reason that $z_{1}$/$Z_{1}$ needs the subscript.
- You should cite the origin of WAIC (I believe this is S. Watanabe 2010 or 2013) not just the Vehtari paper.
- I would like to see error bars on Table 1: you've already run each experiment 16 times, and these are presumably the means? So why not include a standard deviation?
- Appendix Section **C.3 No U-Turn Sampler** should either report some statistic of convergence (split R-hat, IAC, or ESS) for NUTS, or at least report that some minimum threshold of stationarity was reached during each run.
- I don't think there's any use in reporting the "steps to convergence", since a step is so inherently different for each algorithm (at least, it's very different between CAVI/BBVI and CAVI/MLE). You could argue that it's useful to show the scaling behavior with training set size, but this is already better quantified by the wall-clock runtime.

## Minor
- Even though CAVI-CMN is "only" competitive with BBVI and NUTS on WAIC, it would still be nice to bold the best statistic in each column. Then the reader can compare CAVI-CMN at a glance by looking from the top row to the bolded item. This would, in fact, make it clearer that CAVI is competitive, even if it is not consistently the "best".
- I'm sure it's implicitly in the code provided, but at the bottom of page 13 in Appendix Section **D Dataset Descriptions** it might be nice to describe how the random parameters were initialized (i.e. as samples from the priors, or uniformly in some range, etc.).

---

### Decision · Program_Chairs · 2024-10-09

Accept (Poster)